# Si-Wu Water Extracts Protect against Colonic Mucus Barrier Damage by Regulating Muc2 Mucin Expression in Mice Fed a High-Fat Diet

**DOI:** 10.3390/foods11162499

**Published:** 2022-08-18

**Authors:** Zheng Ruan, Yujuan Yu, Peiheng Han, Li Zhang, Zhongyi Hu

**Affiliations:** 1State Key Laboratory of Food Science and Technology, Institute of Nutrition, School of Food Science and Technology, Nanchang University, Nanchang 330047, China; 2Jiangxi Food Inspection and Testing Research Institute, Nanchang 330001, China

**Keywords:** si-wu water extract, high-fat diet, mucus barrier, endoplasmic reticulum stress, O-glycosylation

## Abstract

A high-fat diet (HFD) could cause gut barrier damage. The herbs in si-wu (SW) include dang gui (*Angelica sinensis* (Oliv.) Diels), shu di huang (the processed root of *Rehmannia glutinosa Libosch.*), chuan xiong (rhizome of *Ligusticum chuanxiong* Hort.), and bai shao (the root of *Paeonia lactiflora f.* pilosella (Nakai) Kitag.). Si-wu water extracts (SWE) have been used to treat blood deficiency. Components of one herb from SW have been reported to have anti-inflammatory and anti-obesity activities. However, there have been no reports about the effects of SWE on gut barrier damage. Therefore, the aim of the study was to explore the effect of SWE on gut barrier damage. In this study, we found that SWE effectively controlled body weight, liver weight, and feed efficiency, as well as decreased the serum TC level in HFD-fed mice. Moreover, SWE and rosiglitazone (Ros, positive control) increased the colonic alkaline phosphatase (ALP) level, down-regulated serum pro-inflammatory cytokine levels, and reduced intestinal permeability. In addition, SWE increased goblet cell numbers and mucus layer thickness to strengthen the mucus barrier. After supplementation with SWE and rosiglitazone, the protein expression of CHOP and GRP78 displayed a decrease, which improved the endoplasmic reticulum (ER) stress condition. Meanwhile, the increase in Cosmc and C1GALT1 improved the O-glycosylation process for correct protein folding. These results collectively demonstrated that SWE improved the mucus barrier, focusing on Muc2 mucin expression, in a prolonged high-fat diet, and provides evidence for the potential of SWE in the treatment of intestinal disease-associated mucus barrier damage.

## 1. Introduction

Changes in dietary patterns lead to the consumption of high amounts of sugar and saturated fat. The consumption of a prolonged high-fat diet promotes metabolic disorder (e.g., obesity, diabetes, hypertension, and cardiovascular diseases) [1]. Mounting evidence revealed that a long-term high-fat diet influenced the function of intestinal stem cells and progenitor cells, and the cellular origins of intestinal dysplasia, which are associated with adenomatous growth [2]. Consistent with this pro-inflammatory response to the HFD, epidemiological studies have implicated high dietary fat intake with an increased risk of inflammatory bowel disease [3]. A HFD in mice and obesity in humans appear to diminish intestinal barrier function, as measured by the passage of endotoxins into the circulation [4]. Gulhane et al. supported the fact that HFD-associated endoplasmic reticulum (ER) stress occurs in gut secretory goblet cells, causing an inflammatory response and reducing the synthesis and secretion of mucous proteins in in vitro models [5]. Meanwhile, the effects of a HFD on ER stress of goblet cells and intestinal mucin properties still need to be explored through in vivo models.

The integrity of the gastrointestinal tract is essential for maintaining health. The mucus barrier serves as the first line of innate host defense that effectively protects against harmful substances while allowing nutrients to penetrate through and reach the epithelia [6,7]. Evidence from decades of research has shown that the mucus layer becomes thinner or more porous in colitis patients and mouse models, allowing microorganisms to invade the epithelial cells and cause intestinal damage [8,9,10]. Mucin is a core component of maintaining the integrity of the mucus barrier, production of which is characteristic of mucosal goblet cells. Muc2, the secreted gel-forming mucin, is the major mucin of the colon mucus and is a large glycoprotein characterized by abundant and variable O-glycan [11]. Muc2 is regarded as a marker of colorectal carcinoma, and there is a lot of research about its effects on intestinal diseases [12,13]. Studies have examined the changes in mucus in HFD models [14,15].

Si-wu is composed of four crude herbs, including dang gui (*Angelica sinensis* (Oliv.) Diels), shu di huang (the processed root of *Rehmannia glutinosa Libosch.*), chuan xiong (rhizome of *Ligusticum chuanxiong* Hort.), and bai shao (the root of *Paeonia lactiflora f.* pilosella (Nakai) Kitag.). Si-wu is a blood-building decoction (Chinese Medical Concept: bu-xie) to improve a deficiency of blood (xue-xu) [16,17]. According to the literature [18,19], si-wu extracts were demonstrated to be of several natural product groups, such as phenolic compounds, phthalides, alkaloid, terpene glycoside, iridoid glycoside, and so on. The polyphenols, like gallic acid and ferulic acid had stronger antioxidative effects than other compounds and bai shao is a major contributor [20]. Moreover, a clinical trial offered the conclusion that oral administration of si-wu decoction for 6 months in healthy volunteers decreased serum oxidation and improved the lipid profile [21]. Dang gui and chuan xiong are rich in essential oil, which has been adopted to promote blood circulation and dispel blood stasis [22]. Every single plant in si-wu has also been widely marketed as a health food for women’s care in countries where Asian medicine is used, and si-wu is also beneficial for gastrointestinal health [23,24,25,26]. In addition, the combination of Puerariae Lobatae Radix and Chuanxiong Rhizoma could improve intestinal permeability, and relieve gut microbiota dysbiosis and gut barrier disruption [27]. Huangqin decoction, of which the major active component is paeoniflorin, improved the pathological morphology and mucosal defects of colonic tissue [28]. Although individual substances or active ingredients may focus on the intestinal barrier, whether the combination of the four herbs can improve mucus barrier damage remains unclear. Based on the above research, therefore, we assume that prolonged supplementation with SWE can improve the high-fat diet-induced damage on the mucus barrier by enhancing the mucin Muc2 mass.

## 2. Materials and Methods

### 2.1. Reagents and Materials

Rosiglitazone was purchased from Solarbio (Cat. No. R8470). Hematoxylin, eosin dye (H&E) and alcian blue/periodic acid–Schiff (AB-PAS) dye solution were obtained from Baiqiandu Technology Co., Ltd. (Wuhan, China). Antibodies for Muc2 were obtained from Abcam Inc (Cambridge, MA, USA, GB11344). Antibodies for Ulex europaeus agglutinin IUEA1 were obtained from GeneTex (GTX01512). Commercial kits were purchased from Nanjing Jiancheng Bioengineering Institute: TC (A111-1-1) and TG (A110-1-1). Mice ELISA kits were purchased from Jiangsu Enzyme industry Co., Ltd. (Taizhou, China): DAO (MM-0228M1), D-La (MM-43853M1), LPS (MM-0634M1), TNF-α (MM-0132M1), IL-6 (MM-0163M1), IL-1β (MM-0040M1). Monoclonal antibodies: anti-CHOP (L63F7, CST), anti-GRP78 (11587-1-AP, ProteinTech Group, Inc., Wuhan, China), anti-Muc2 (ab272692, Abcam), and β-actin (20536-1-AP, ProteinTech Group, Inc.). Polyclonal antibodies: Cosmc (19254-1-AP, ProteinTech Group, Inc.) and C1GALT1 (ab230837, Abcam).

### 2.2. Composition and Preparation of SWE

The preparation of the SWE was based on the method described by Lee et al., with some modifications [29]. The four herbs were purchased from Changsheng pharmacy (Nanchang,, China) and were identified by professor Lan Cao from the Research Center of Chinese Medicine Resources, Jiangxi University of Traditional Chinese Medicine (Nanchang, China). Four herbs were mixed and pulverized at the same weight ratio of 50 g each, immersed in an 8-fold amount of distilled water and boiled in a porcelain pot, until half of the original amount of water was left. The concentrated fluid was then filtered to produce the final water extract (0.25 g/mL). After cooling, we divided the water solution into 15 mL centrifuge tubes, and stored them at −20 °C until use. The extraction was performed using Waters ACQUITYTM UPLC BEH C18 analytical columns with column size 2.1 mm × 50 mm and particle size 1.7 μm (Waters Corporation; Milford, MA, USA). The separation was performed on the following gradient program: 50–70% B at 2–8min, 70–70% B at 8–11 min, 70–90% B at 11–13 min, 90–90% B at 13–15 min, 90–15% B at 15–15.01min. The optimal MS parameters were set as follows: capillary voltage 3 kV, desolvation gas flow rate 600 L/h at a temperature of 500 °C, cone gas flow rate 50 L/h, and source temperature 120 °C. The mass spectrometry was programmed to perform in both positive and negative ionization modes and under the full-scan analysis with mass range of *m*/*z* 50–1500.

### 2.3. Animals and Experimental Protocols

Animal studies were approved by the Institutional Animal Ethical Committee, Nanchang University (permission number: SYXK2015-0001).

Healthy six-week-old male C57BL/6J mice were purchased from SJA Laboratory Animal Co., Ltd. (Changsha, China) and housed in a specific pathogen-free facility at 23 °C with a 12/12 h light−dark cycle. After acclimation for one week, the mice were randomly assigned to four groups (n = 8 for each group, Figure 1A). The control group (CON) received a standard diet for 12 weeks. The standard diet contained approximately 10% kcal from fat (Cat #D12450J, New Brunswick, NJ, USA). The HFD group received a high-fat diet for 12 weeks, containing approximately 60% kcal from fat (mixed lard and soybean oil; Cat #D12492, New Brunswick, USA. Diet composition is described in Appendix A). The SWE group received a HFD and 10 mL kg^−^^1^/d SWE gavaged for 12 weeks. The rosiglitazone group (POS) received a HFD for 12 weeks and was administered rosiglitazone (20 mg/kg BW) daily by gavage at the 11th and 12th weeks. The dose of rosiglitazone was based on results of the previous study, which demonstrated that 20 mg kg^−^^1^/d promoted intestinal barrier function by improving mucus and tight junctions in a mouse colitis model [30]. rosiglitazone treatment for two weeks is to avoid significant toxicity from a prolonged administration at this dose [31]. Mice in the CON and HFD groups were gavaged with an equal volume of normal saline. Body weight and food intake were recorded weekly during the study to analyze the effect of SWE. All samples were obtained at the end of the 12 weeks of the experimental period.

### 2.4. Determination of the Total Phenolic and Flavonoid Contents

The determination of the total phenolic content was performed by the Folin–Ciocalteu method [32]. Briefly, samples were mixed with the Folin–Ciocalteu reagent and incubated for 5 min, and then 7.5% Na_2_CO_3_ solution was added to the mixture. The absorbance was measured at 765 nm after one hour incubation at room temperature. The standard working solution of Gallic acid (concentration 0–1000 μg/mL) was configured to prepare the standard curve. The total phenolic content was expressed as mg gallic acid equivalents (GAE) per mL of extract.

The total flavonoid content of the samples was detected according to the method described by Yang et al. with some modifications [33]. Samples were mixed with 5% NaNO_2_ solution and incubated for 5 min, then a 10% Al (NO_3_)_3_ solution was added and incubated for another 6 min. The reaction was terminated by adding a 1 mmol/L NaOH solution, and the absorbance of the mixture was measured after 10 min incubation at 500 nm. The standard working solution of rutin was configured to prepare the standard curve. The total flavonoid content was expressed as mg rutin equivalents (RE) per mL of extract.

### 2.5. Inflammatory Cytokines and Serum Endotoxin Detection

The serum was obtained after centrifugation at 3500 rpm at 4 °C for 15 min. We measured the serum parameters (TC, TG) with commercial kits. DAO, D-La, and LPS in the serum are commonly regarded as indirect indicators of intestinal permeability, and the concentrations were determined using ELISA kits. The concentrations of TNF-α, IL-6, and IL-1β in serum, and ALP levels in colon tissue were measured using commercial ELISA kits.

### 2.6. Histologic and Immunohistochemical Analyses

**H&E staining:** After the mice were killed, colon sections were fixed in 4% paraformaldehyde over 24 h at room temperature. The samples were then paraffin-embedded and cut into 4 μm transverse sections for conventional H&E, as described [34]. Finally, we could observe that the nucleus was blue and cytoplasm was red via light microscope (Nikon Eclipse E100). Morphological changes were examined and the histological score was measured based on the work of Murano et al. [35].

**AB-PAS:** To identify the mucus layer thickness and goblet cells, tissue sections were stained with AB-PAS according to the manufacturer’s instructions. AB-PAS sections were scanned by Pannoramic Scan (3DHISTECH, Hungary) to obtain the staining pictures. Mucus thickness refers to the thickness of the mucus layer attached to the surface of the colonic epithelial cells. For the determination of the mucus layer thickness, 3 to 6 representative slices were selected from each group, and each slice was measured at five different areas. Goblet cells were counted according to the general morphological characteristics per crypt. The mucus layer thickness and goblet cells were measured using AB-PAS staining pictures with CaseViewer 2.4.0.

**Immunofluorescence:** According to the previous research [36], some changes were made to the staining method that was used for fixation of the colon tissues, followed by embedment in paraffin, and finally, immunofluorescence staining was performed. Sections were observed under a fluorescence microscope (Nikon Eclipse C1) and images were collected (Nikon DS-U3). Quantitative analysis of immunofluorescence staining was measured by CaseViewer (3DHISTECH, Budapest, Hungary) and ImageJ (Media Cybernetics, Rockville, MD, USA).

### 2.7. Protein Quantitative Analysis

We used a Western blot method similar to that used by Qiu et al., with some modifications [37]. The total protein from the colon tissues was extracted with RIPA lysis buffer (Beyotime, Peking, China). The protein samples were loaded onto SDS-PAGE. Protein bands were detected with an ECL kit (Beyotime, Peking, China) and visualized using ChemiDoc MP System (Bio-Rad Laboratories, Inc., Hercules, CA, USA). Finally, the signal plots were processed with ImageJ 1.8.0 software.

**ELISA analysis:** C1GALT1 and Cosmc protein quantitation of the colon were detected using commercial ELISA kits. All experimental procedures were performed according to the manufacturers’ instructions.

### 2.8. Statistical Analysis

Statistical analysis was driven as mean ± standard error of mean via Statistical Package for Social Sciences (SPSS) software (IBM SPSS Statistics for Windows, IBM Corp. and Version 26.0.). One-way analysis of variance (ANOVA) was used for multiple comparisons to evaluate the statistical significance. The results were expressed relative to the value recorded in the CON group as equivalent to 1. Correlations were analyzed using the Pearson test. A *p* value < 0.05 was considered significant and a *p* value < 0.001 was considered highly significant. All graphical representations were made using GraphPad Prism 8.0.2 software.

## 3. Results

### 3.1. Determination of Total Phenolic and Flavonoid Contents

The levels of phenolic and flavonoid compounds were quantitatively measured to help us elucidate SWE’s potential protective mechanism against mucus barrier damage. The results showed the total phenolic and flavonoid contents were 4.10 mg GAE/mL and 2.63 mg RE/mL, respectively. Appendix A showed that UPLC-Q/TOF-MS analyzed SWE’s typical total ion chromatogram in positive ion mode, and identified major active compounds in terms of the MS data in previous literature, including aromatic acids, flavones, polysaccharides, monoterpene glycosides, and others [38]. Paeoniflorin and (E)-ligustilide exerted high response values, and chlorogenic acid was found to be the primary phenolic acid in the SWE, which was derived mostly from dang gui and chuan xiong [39].

### 3.2. SWE Reduced the Signs of Obesity after High-Fat Diet Intervention

As shown in Figure 1B, all groups had similar initial body weight (21.44–21.71 g). From the 9th week onwards, the HFD group, identical to the POS group, showed a significant difference in the final body weight and feed efficiency (Figure 1D) in the CON and SWE groups. Food intakes, measured weekly, were similar across all groups (Figure 1C). At the end of the 12th week, feeding a high-fat diet caused liver weight gain (Figure 1E) and increased epididymis fat accumulation (Figure 1F), which significantly decreased after SWE supplementation, with no difference in the POS group. As showed in Figure 1G, the average of serum TC level was 3.355 mmol/L in the CON group, 6.895 mmol/L in the HFD group, 5.570 mmol/L in the SWE group, and 5.173 mmol/L in the POS group. Clearly, the HFD group was twice as high as the CON group. The SWE group decreased by 19.2% and the POS group decreased by 24.97% compared with the HFD group. As for the serum TG levels (Figure 1G), the CON and POS groups showed a significant difference compared with the HFD group. Nevertheless, changes in the TG level between the HFD and SWE groups were not significant.

### 3.3. Effects of SWE on Colonic Mucosal Morphology, Intestinal Permeability, and Inflammation

The H&E staining results revealed that the colon tissue structure was typical in the CON group (Figure 2Aa). The HFD-fed mice exhibited considerable infiltration of inflammatory cells, which destroyed mucosal integrity (Figure 2Ab). Moreover, a significantly reduced influx of inflammatory cells was observed in the colon of the SWE (Figure 2Ac), and rosiglitazone-treated mice (Figure 2Ad). The morphology scores to quantify abnormalities of colon pathology (Figure 2B) were consistent with those of H&E staining. Compared with the HFD group, the level of colonic ALP was significantly increased in SWE supplemented mice (Figure 2C). However, the serum LPS level was not markedly decreased following the ALP level increase (Figure 2D). D-La (Figure 2E) and DAO (Figure 2F) levels were notably decreased in the SWE and POS group compared with the HFD. Compared to the control group, the TNF-α level was increased in the HFD group and SWE group. Meanwhile, the TNF-α level in the SWE group was 13% higher than in the HFD group (Figure 2G), while the levels of IL-6 (Figure 2H) and IL-1β (Figure 2I) were significantly decreased compared with those of the HFD group.

### 3.4. Effects of SWE on Colonic Mucus Layer Thickness and the Number of Goblet Cells in HFD-Fed Mice

The number of goblet cells and mucus layer thickness were investigated with AB-PAS staining (Figure 3A). The lower mucus thickness was detected in HFD-fed mice (Figure 3B) compared with the CON group. Following SWE and rosiglitazone treatment, the mucus layer thickness was significantly increased. Consistent with the variation trend of mucus layer thickness, the number of goblet cells (Figure 3C) was observed to significantly increase in the CON, SWE, and POS groups compared to the HFD group.

### 3.5. Effects of SWE on the Production and Distribution of Colonic Mucin

Immunofluorescent staining revealed that the average optical degree (AOD) of Muc2 in the HFD group was decreased. In contrast, an increase in AOD of Muc2 in the other groups was observed (Figure 4A,B). We then tested whether the increase of Muc2 secretion was accompanied by appreciable changes in O-glycosylation. We used lectin UEA1 to monitor fucose on O-glycan. The AOD demonstrated that mucin-special O-glycans were specifically increased after SWE supplementation or rosiglitazone administration (Figure 4C,D). Similar trends were observed when Muc2 protein was evaluated by western blot (Figure 4E,F). Collectively, our data illustrate that SWE supplement could restore the function of the mucus barrier in HFD-fed mice.

### 3.6. Effects of SWE on Colonic Mucin Procession

We determined the ER stress chaperone GRP78 (glucose-regulated protein 78) and apoptosis marker CHOP (C/EBP homologous protein) with western blot to examine the contribution of ER stress. In high-fat feeding, the expression of the proteins of GRP78 and CHOP was over 30% higher than in the SWE and POS groups (Figure 5A–C), which suggested that SWE or rosiglitazone treatment significantly inhibited ER stress. To further confirm the protective effects of SWE, we detected the relevant protein expression T-synthase (also known as C1GALT1, core 1 synthase, glycoprotein-N-acetylgalatosamine3-galactosyltransferase) and its unique molecular chaperone, Cosmc (also known as C1GALT1C1, core 1 β1, 3-galactosyltransferase–specific chaperone 1) associated with core 1-derived glycans. In western blot analysis (Figure 5D–F), the C1GALT1 and Cosmc protein expression levels in the CON and SWE groups were significantly higher than in the HFD group, consistent with results from the ELISA analysis (Figure 5G,H). In the POS group, there was no difference found on the Cosmc protein level but we observed a significant increase in ELISA analysis compared with the HFD group.

## 4. Discussion

SWE contains a variety of bioactive compounds which have an important role in health. A previous study established HPLC fingerprinting and LC-MS quantification methods for SWE and determined multiple active ingredients [40]. In the present study, we detected the components by UPLC-Q/TOF-MS and identified compounds with relatively high responses, including (E)-ligustilide, chlorogenic acid, and verbascoside (14.78%, 13.77%, and 11.58%, respectively, based on the percentage of paeoniflorin) (Appendix A). There have been many reports in the literature that these bioactive components exert beneficial effects on gut health [24,41,42,43,44,45,46]. Si-wu administration before irradiation protected the jejunal crypts [47]. Our results indicate that SWE improves the mucus barrier by regulating the mucin Muc2. Indeed, we observed that the increase in colonic goblet cell numbers in the SWE group reflects the increased ability of SWE to promote secretion of Muc2. Particularly, the morphology, immunofluorescence, and western blot testing all confirmed an increase of Muc2 content.

On the other hand, abnormal mucin O-glycosylation was recovered by improving the expression of the key rate-limiting enzyme (C1GALT1) and increasing the density of UEA-1 that monitors fucose on O-glycan. Therefore, it is possible that SWE improves the mucus barrier by regulating the mucin muc2 in terms of secretion capacity, secretion amount, and glycosylation processing.

Emerging evidence has proven that HFD can induce intestinal barrier damage through enhancing intestinal permeability associated with inflammation and gut bacteria dysbiosis, altering mucus properties, disrupting tight junctions and intestinal epithelial shedding–proliferation axis, and so on [48]. Not surprisingly, our results showed that SWE effectively improved HFD-induced mucus barrier damage. SWE maintained the number of goblet cells, increased Muc2 production and secretion, and enhanced the mucus layer thickness. In particular, systemic inflammation and intestinal permeability were improved, reflected by the increasing ALP level, the decrease in the content of inflammatory factors (IL-6 and IL-1β) as well as the DAO and D-La levels. An increase in TNF-α level in the SWE group was detected in the present research, which may suggest that SWE promotes the body’s immune response by inducing the macrophages to release TNF-α. It is reported that the four SWE constituents increase the colony-forming granulocyte–macrophage unit [49]. Similar to TNF-α, regarding the biological effects, IL-6 not only induces signs of acute inflammation but also elicits anti-inflammatory effects [50,51]. The connection of these inflammatory factors (TNF-α, IL-6, and IL-1β) is not only unidirectional, but also has multiple signaling pathways. Ceramide kinase (CERK) is expressed at higher levels in obese individuals, and by inhibiting CERK expression, TNF-α -induced IL-1β secretion is reduced [52]. Additionally, both TNF-α and IL-1β are sufficient to induce IL-6 promoter activity, both signaling pathways are required for IL-6 active transcription. Different from TNF-α, IL-1β may generate a temporal binding of C/EBPβ to NF-IL-6 consensus to induce the secretion of IL-6 [53]. Besides, TNF-α-induced IL-1β production was significantly blocked by inhibition of long-chain acyl-CoA synthetase 1 (ACSL-1) activity [3]. Taken together, although elevated TNF-α was observed in SWE, it may inhibit IL-6 and IL-1β secretion through other signaling pathways such as ACSL-1-mediated or C/EBPβ to NF-IL-6 signaling.

DAO is an intracellular enzyme in intestinal mucosa. D-La is the product of the degradation and cleavage of the bacteria inherent in the gastrointestinal tract, such as *Lactobacillus*. The tissue of lactating animals cannot produce or slow down D-La. When the intestinal mucosal barrier is damaged, D-La can penetrate through the intestinal mucous membrane into the blood. The accumulation of D-La and DAO can increase the permeability of the intestinal membrane. Therefore, the measurement of serum DAO and D-La activity or concentration can be an easy and convenient modality for evaluating intestinal permeability [54]. In the present study, the levels of D-La (Figure 2E) and DAO (Figure 2F) were notably increased in the HFD group but significantly decreased with the administration of SWE. Muc2 is the main component of the intestinal mucus barrier and is secreted by intestinal epithelial goblet cells. In our experiment, the observed accumulation of D-La and DAO levels in blood reflects the damage of intestinal epithelial cells, including goblet cells. This is consistent with our findings (Figure 3C). This may be one of the reasons why the mucus barrier is compromised. However, the serum LPS level did not reverse with SWE supplementation. Further study is needed to evaluate intestinal permeability using more specific and invasive markers (Dextran-FITC, mannitol, etc.) of different sizes.

In this study, we explored the quantitative changes in mucin secretion, as well as structural changes in mucin’s glycoprotein core following supplementation with SWE in HFD-fed mice. As a covering layer on the intestinal epithelium, mucus is composed of mucin rich in O-glycosylation. The mucus barrier is the first line of defense to protect the gastrointestinal tract. A high-fat diet weakens the intestinal mucus layer [48,55]. The thin mucus layer was characterized by goblet cell depletion, low expression of Muc2, and low abundance of bacteria, which led to a series of unhealthy reactions [56,57]. The dysregulation of key enzymes involved in the O-glycosylation pathway leads to the presence of new glycanic receptors at the mucin surface. These receptors mask cancer cells from the immunologic system and favor the development of tumor at a secondary site [58,59]. The increased number of goblet cells and the recovery of key enzyme CIGALT1 and its chaperone Cosmc promote MUC2 protein expression, in the present experiment. In addition, cellular inflammatory factors also affect mucin gene expression. Levine et al. found that the exposure of TNF-α induced the increase in MUC2 steady-state mRNA levels in human airway epithelial cells [60]. Is there a similar situation in the intestinal barrier? Iwashita et al. showed that exposure of TNF-α can induce the up-regulation of MUC2 mRNA through a pathway other than the mitogen-activated protein kinase pathway in the human colonic cancer cell line LS174T [61]. Further validation in animals is required, because elevated pro-inflammatory cytokines also trigger the body’s inflammatory response.

Impaired expression of Muc2 O-glycan and ER stress has recently been reported to lead to disruption of the colonic mucus barrier, colitis, and related cancers [62,63,64]. Our study showed that the SWE enhanced the goblet cell numbers and Muc2 level of protein expression. Here we also showed that SWE can mitigate ER stress in colon secretory goblet cells and the unfolded protein response (UPR). The UPR is correlated with Muc2 protein expression. Moreover, the improvement in systemic inflammation and gut permeability maintain the mucus mass, in this paper. These data reflect the multi-targeted effect of SWE on improving body health. It is reported that Shaoyao decoction enhanced the Muc2 levels in the colon of the DSS-induced colitis model, and *Angelica sinensis* protected H9C2 cells against ER stress by activating the ATF6 pathway [65,66]. Therefore, one of the reasons that SWE improved the mucus barrier may be thanks to the active ingredients in *Angelica sinensis* and Radix paeoniae alba. Therefore, ER stress interferes with protein folding and interrupts the Muc2 O-glycosylation process, resulting in reduced mucin secretion and a thinner mucus layer [67]. What are the specific pathways that link ER stress to O-glycosylation? Ju et al. found that GRP78 can co-immuno-precipitate with inactive T-synthase in the absence of Cosmc [68], which suggested a specific interaction between GRP78 and T-synthase. It is also reported that Cosmc overexpression, which is not correlated with elevated T-synthase, may be induced by ER stress [69].

We found that the barrier-protective effect of the SWE was mediated through the mass and modification of mucin. The findings in the present study provide evidence that SWE has a potential protective effect on intestinal barrier function. Further studies are in progress to establish the mechanism underlying the role of SWE on mucin production and secretion in the colon.

## 5. Conclusions

In summary, we demonstrated that SWE in HFD-fed mice efficiently improved mucus barrier damage and mitigated colon diseases. Long-term supplementation with SWE reduced body weight and bloods lipid in HFD-fed mice. The protective effects also contributed to an increase in the level of ALP, down-regulated the inflammatory cytokines (IL-6, and IL-1β), and decreased intestinal permeability (DAO, and D-La), but SWE did not alter the serum LPS level. In addition, SWE supplement mitigated ER stress, improved the O-glycosylation process, and maintained the secretion of mucin. These findings provide a foundation for developing si-wu to cure intestinal disease associated with mucus barrier damage.

## Figures and Tables

**Figure 1 foods-11-02499-f001:**
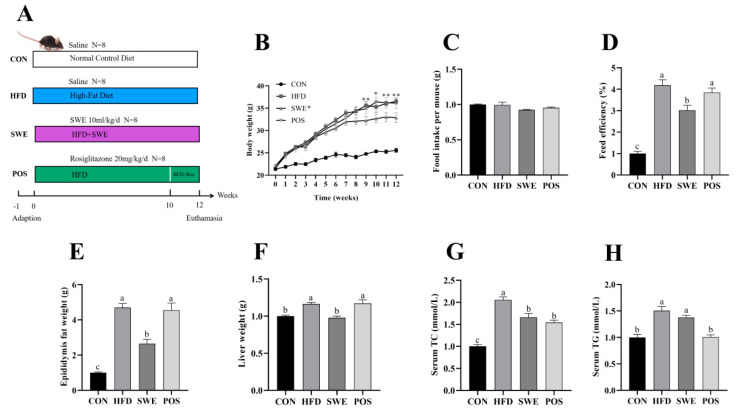
Effects of SWE on growth performance in the HFD−fed mice. (**A**) schema showing the animal groups and treatments, (**B**) body weight change, (**C**) food intake, (**D**) feed efficiency (body weight gain/food intake), (**E**) epididymis fat weight, (**F**) liver weight, (**G**) serum TC level, (**H**) serum TG level. Values are expressed as mean ± S.E.M. (* *p* < 0.05, ** *p* < 0.001, vs. SWE). Means marked with superscript letters are significantly different to the others (*p* < 0.05). n = 6−8 per group.

**Figure 2 foods-11-02499-f002:**
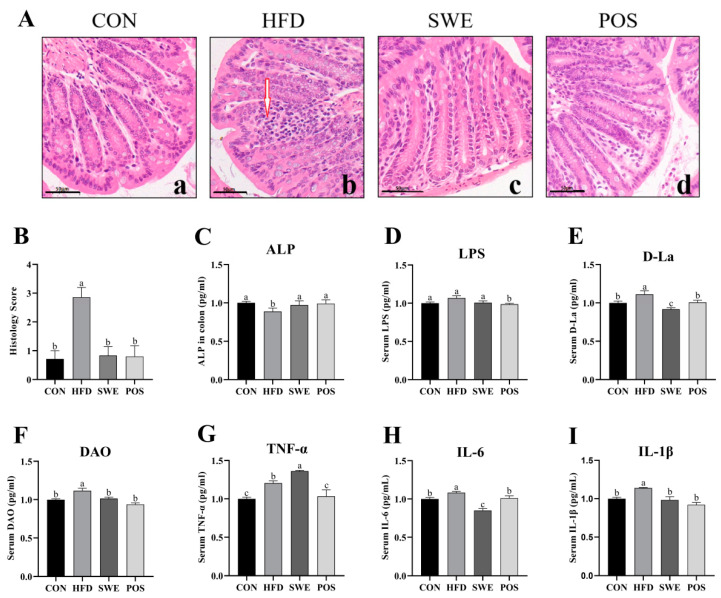
Effects of SWE on colonic mucosal morphology, intestinal permeability, and inflammation. (**A**) representative photographs of the distal colon sections with H&E staining (400×). Scale bar: 50 μm; red arrow indicates inflammatory infiltration. The a, b, c, d in subgraph A represent the order of the pictures for easy elaboration, (**B**) Histological score of the colon tissues (n = 5–7), (**C**) ALP level in colon, (**D**) LPS level, (**E**) D-La level, (**F**) DAO level, (**G**) serum TNF-α level. (**H**) serum IL-6 level, (**I**) serum IL-1β level. Values are expressed as mean ± S.E.M. Data with different superscript letters are significantly different (*p* < 0.05). n = 4–8 per group.

**Figure 3 foods-11-02499-f003:**
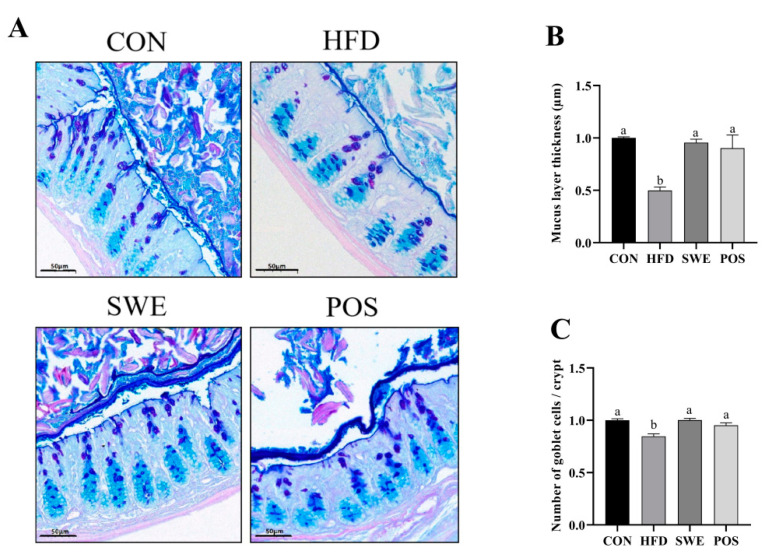
Effects of SWE on colonic mucus thickness and number of goblet cells in HFD-fed mice. (**A**) representative AB/PAS-stained distal colon sections showing mucus layer thickness and goblet cells within the colon tissue, scale bar: 50 μm, (**B**) manually measured colonic mucus layer thickness, (**C**) number of goblet cells per crypt (5–8 sections per animal). Values are expressed as mean ± S.E.M. Data with different superscript letters are significantly different (*p* < 0.05). n = 4–6 per group.

**Figure 4 foods-11-02499-f004:**
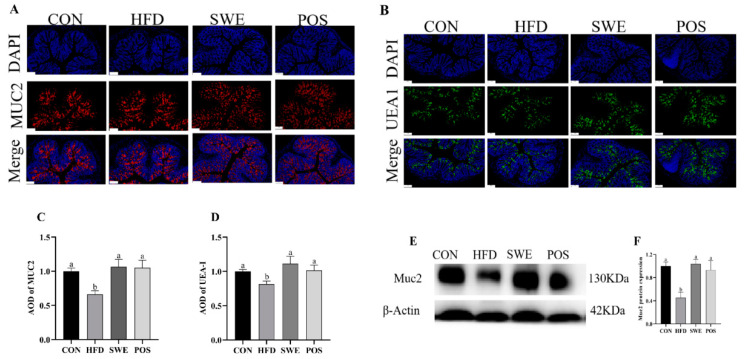
Effects of SWE on the content and distribution of colonic mucin. (**A**) representative immunofluorescent staining in the distal colon using an antibody against Muc2 (red) with DAPI (blue). Scale bar: 100 μm, magnification 200×, (**B**) fluorescent images showing O-glycans by using UEA1 staining. Scale bar: 100 μm, magnification 200×, (**C**) average optical density (AOD) of Muc2, (**D**) average optical density of UEA1, (**E**) Muc2 protein band diagrams, (**F**) Muc2 protein expression. Values are expressed as mean ± S.E.M. Means marked with superscript letters are significantly different to the others (*p* < 0.05). n = 4–8 per group.

**Figure 5 foods-11-02499-f005:**
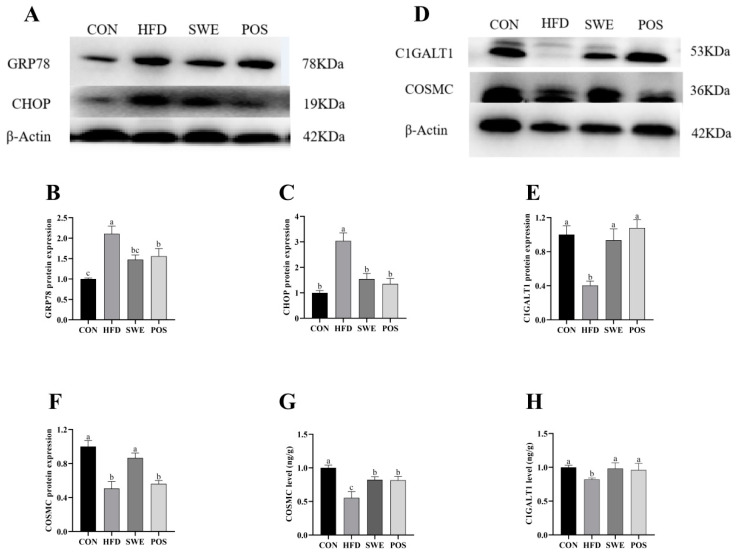
Effects of SWE on colonic ER stress markers and mucin O-glycosylation in the colon. Western blot analysis: (**A**) protein abundance of GRP78 and CHOP, (**B**) GRP78 protein expression, (**C**) CHOP protein expression, (**D**) protein abundance of C1GALT1 and Cosmc, (**E**) C1GALT1 protein expression. (**F**) Cosmc protein expression. ELISA analysis: (**G**) C1GALT1 level in colon tissue, (**H**) Cosmc level in colon tissue. Values are expressed as mean ± S.E.M. Different letters indicate significant differences (*p* < 0.05). n = 4–8 per group.

## Data Availability

All data presented in this research are available through the corresponding author.

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
