# Peer review of "Si-Wu Water Extracts Protect against Colonic Mucus Barrier Damage by Regulating Muc2 Mucin Expression in Mice Fed a High-Fat Diet"

_foods, 2022, doi:10.3390/foods11162499_

Round 1
Reviewer 1 Report
The authors studied that the effects of herbs of Si-Wu in HFD induced gut barrier damage. They found Si-Wu water extracts (SWE) efficiently improved the mucus barrier damage and mitigated colon diseases. They also found the level of ALP was increased, the inflammatory cytokines (IL-6, and IL- 1β) and the intestinal permeability (DAO, and D-La) were decreased after the use of SWE, but the serum LPS level was not changed. SWE increased the goblet cell numbers and mucus layer thickness. In addition, SWE supplement mitigated the ER stress, improved O-glycosylation process and maintained the secretion of mucin. These findings provide a foundation for developing the Si-Wu to cure intestinal disease associated with mucus barrier damage.
This is a very interesting study. However, some problems need to be more clarified.
The title: Si-Wu water extracts protect against colonic mucus barrier damage by regulating Muc2 mucin expression in High-Fat Diet mice
My opinion is: Although Si-Wu water extracts protect against colonic mucus barrier damage, and it also regulates Muc2 mucin expression in High-Fat Diet mice. However, no solid data proved that the protective function is through the muc2 mucin expression in this experiment.
Other comments:
Major
1. the mice received HFD for 10 weeks and in a dose of 20 mg kg-1/d rosiglitazone intragastrically in the last two weeks of the experiment.=> received HFD for 12 weeks? Please clarify
2. Rosiglitazone treatment for two weeks is to avoid a significant toxicity in a prolonged administration at this dose Does Si-Wu have significant toxicity in a prolonged administration, why no control + SWE group? Why no studying the possible toxicity?
3. Serum total TC (Figure 1G) and TG (Figure 1H) levels were 30% and 60% higher, respectively, in the HFD group compared with CON and POS group. Please clarify this sentence. Serum total TC (Figure 1G) levels were 30% higher in the HFD group compared with CON and POS group, why? According to the figure, the level in the control is about 3.+ and the level is about 7 in the HFD group. Why only 30% higher?
4. The number of goblet cells in the crypt (Figure 3C) was 2-fold higher in other three groups (CON, SWE and POS) than the HFD group. According to the figure, it seemed not reach 2-fold.
5. in HFD mice, we observed two-fold lower levels of C1GALT1 protein expression than CON and SWE group. Please clarify this sentence. Half seemed better than 2-fold lower.
Minor:
1. Changes in dietary patterns favor the consumption of high amounts of sugar and saturated fat. What’s the meaning of “favor”
2. Does the combination of the four component herbs in Si-Wu have fixed ratio?
3. The concentrate fluid was then filtered and got the final water extract (0.25 g/mL): the density is so low, please recheck
Author Response
Thank you for your constructive comments on our article (Manuscript ID: foods-1804337). These comments are valuable and helpful in improving our articles. All of these comments are carefully discussed by all authors. Based on the comments, we have done our best to revise the manuscript to meet the journal's requirements.

Reviewer 2 Report
The study analyzed the effect Si-Wu water extracts on gut barrier damage in mice fed a HFD. I have some issues that need to be addressed.
Introduction
Lines 58-59, Si-wu extract, is formed by four herbs? Please, clarify this information.
The authors need better describe the properties of the Si-Wu extract. It isn’t very clear for non-Chinese readers. Do early studies characterize the Si-Wu extract properties (polyphenol, monoterpene glycosides, polysaccharides)? In methods, this became clearer. Please, also make it clear in the introduction.
Methods
The authors used SPPS for statistical analyses, but it seems that the graphics were made in another program. Please, if I’m correct, clarify this in the manuscript.
Lines 175-176: The authors described correlation analyses but did not present these results in the manuscript.
Lines 176-177: How to identify this in the figures?
Results:
Lines 205-206: In the footers of the figures, the authors inserted “*p<0.05, **p<0.01”, but this is not shown in the figure. Please, correct it. In addition, the authors describe that “n=6-8” per group was used, but in methods, the authors informed that n=8 animals/group. Please, correct it.
Figure 2: How do the authors explain an increase in TNF-a and reduction in IL1b once IL1 is induced mainly by TNF-a, IFN-a, and LPS? The authors did not describe it in the discussion section.
Lines 226-227. Considering that the authors used varying numbers of mice in experiments, the authors should display the scatter plots for all figures, identifying the results of each mice.
Figure 4. The merged figure of the HFD group seems not to be a reliable figure of DAPI and MUC 2 in HFD. Please, correct and re-analyze it.
Figure 5. In fig 5D, there is two evident protein band in COSMC expression. Please, identify the analyzed band.
Minor
Native English could revise the manuscript.
Lines 60-62, please insert references to this sentence.
Lines 128: Correct the verb.
Lines 141: Levels
Please, clarify “a”, “b”, and “c” in the footers of the figures.
Author Response
we would like to thank you for your constructive comments concerning our article (Manuscript ID: foods-1804337).These comments are all valuable and helpful for improving our article. All the authors have seriously discussed about all these comments. According to the comments, we have tried best to modify our manuscript to meet with the requirements of the journal.

Reviewer 3 Report
Manuscript number 1804337
In the current manuscript titled "Si-Wu water extracts protect against colonic mucus barrier damage by regulating Muc2 mucin expression in High-Fat Diet mice", the authors have tried to establish the role of Si-Wu water extract (SWE) in gut protective function by regulating the expression of mucin (Muc2) and ER stress in a high-fat diet-mediated gut dysbiotic mouse model. This is an interesting piece of work that suggests the protective function of combined herbal extracts in the management of gut barrier dysregulation and colon disease. This study will strengthen the knowledge of therapeutic drugs in the management of colon disease. However, I have the following comments for this study:
Comments
1) According to the authors, Si-Wu is a combination of four medicinal herbs that have significant anti-inflammatory and protective properties in gut-related diseases. Then what are the reasons for using the combined effect?
2) Does combined herb extract (SWE) have a better effect than an individual herb? In this case, the authors should do one experiment with an induvial and a combined dose. How have the authors decided on the dose of SWE?
3) Why do the authors use SWE intragastrically? Is it the same as oral gavage? If not, the authors should explain this in the manuscript to the reader, so they understand.
4) In figure 1, I realized major flaws where I did not understand the meaning of the small alphabet letters on the bar graph (a, b, c, d). It is not mentioned anywhere if I have not missed it. Instead of this, authors should use the p-value or star symbol (*, **).
5) Why does POS control work better in some places but not in other experiments (Figure 1B-F and Figure 1G-H)? Or Figure 2.
6) The high-fat diet model has been linked to gut dysbiosis. Gut dysbiosis is associated with barrier dysfunction and thinning of the mucus layer. The authors found a reduction of IL6 and IL-1beta but not TNF alpha (which increased) and serum LPS. These results suggest that SWE does not affect gut dysbiosis. This is contrasting data.
7) There is no gut dysbiosis-related data. There is no inflammatory data such as NFkB or STAT3. I do not see any novelty in this paper. It is just an observational study.
8) What is the target of this extract? Does it target the liver, the gut, or both because it affects enzymes and genes from the gut and liver?
9) In Figure 4, it seems that the intensity of MUC2 staining in the HFD panel is greater as compared to SWE and Pos. Please give us a better representative.
10) Alcian blue is the staining of neutral mucin, which is a lower firm mucus layer (towards lamina propria). However, the upper (towards lumen) microbial-interaction layer has loose neutral mucin stained by PAS staining. Please perform PAS staining to determine the reduction of the mucus layer.
(11) For all figures, please give the P-value. Please represent the bar graph with an induvial number in the graph.
12) Represent the data in terms of relative to the control (for example, control - 1.0 vs treatment. X).
13) use bold heading for H&E staining, AB-PAS, Immunoflurescence (Line 143-160)
Author Response

(The authors gave the same response as above.)

Round 2
Reviewer 1 Report
I found the units of y-axis in all figures have been changed in version 2. In fact, I prefer the usage of version 1. If the authors like to use the new ones, I think it is better to add ratio in the unit of y axis.
The author had answered the questions. I have no further questions.
Author Response
We would like to thank you for your constructive comments concerning our article (Manuscript ID: foods-1804337). These comments are all valuable and helpful for improving our article. All the authors have seriously discussed about all these comments. According to the comments, we have tried best to modify our manuscript. Point-by-point responses are listed Detailed Response to Reviewers.

Reviewer 2 Report
After review, I have no further comments.
Author Response
Thank you!
Reviewer 3 Report
In this revised manuscript authors made a major improvement to the manuscript and incorporated a reviewer’s comments. However, I have the following comments
1) In my previous comments, I had asked the authors to give a rationale behind the use of Si-Wu (mixture of 4 herbs plan extract. Authors were asked to do a comparative analysis between Si-Wu extract with individual herbal components to determine the efficacy of mixture extract as compared to the individual component (how much fold change). Although authors have tried to give an explanation by taking examples from other plan mixture extracts (Si-WuTang) but not the one used in this study (Combined treatment of gallic acid, Z-liguistilide, and senkyunolide A, showed enhanced effect against JB6 transformation compared with that of the single compound alone [9]. This will be surprising if individual components have some beneficial effect and combined extract will not. Any combined mixture will have a better effect than an individual component. Thus, this explanation and examples are not relevant and satisfactory to the current study.
2) In figure 3 authors have explained that they have done AB-PAS staining to determine the mucus thickness and mucin layer. However, they have not mentioned whether they have used AB or PAS, or both to measure mucus thickness. In their staining, they have not mention which staining method was used for quantification (figure 3A-B). Authors should explain if they can.
3) In Figures 4A and 4C, the staining image and densitometry are not matching for the high fat-diet group for the MUC2. Staining looks brighter than control, but densitometry data shows reduced MUC2 in the HFD group. This was also the question of the previous review. Authors should use a different image if have it.
4) For figures 5D and 5E and 5F, beta-actin loading is not equal. Reduced expression of C1GALT1 and Cosmc proteins could be due to reduced expression of beta-actin in the HFD group. Authors should use better representatives. There are two bands for C1GALT1 and Cosmc proteins. Which brand is the real band for the protein of interest? Authors should point out the expected band.
Author Response

(The authors gave the same response as above.)
